# Risks Associated with the Use of Intranasal Corticosteroids: An Analysis of Adverse Reactions Reported to EudraVigilance

**DOI:** 10.3390/healthcare12191923

**Published:** 2024-09-25

**Authors:** Luís Martins, Samuel Silvestre, Cristina Monteiro

**Affiliations:** 1Faculdade de Ciências da Saúde, Universidade da Beira Interior, 6200-506 Covilhã, Portugal; luisloureiro.loureiro1@gmail.com; 2CICS-UBI-Health Sciences Research Centre, Faculdade de Ciências da Saúde, Universidade da Beira Interior, 6200-506 Covilhã, Portugal; sms@ubi.pt; 3UFBI-Pharmacovigilance Unit of Beira Interior, Faculdade de Ciências da Saúde, Universidade da Beira Interior, 6200-506 Covilhã, Portugal; 4Academic Clinical Center of Beiras, Faculdade de Ciências da Saúde, Universidade da Beira Interior, 6200-506 Covilhã, Portugal

**Keywords:** intranasal corticosteroids, pharmacovigilance, adverse drug reactions, EudraVigilance

## Abstract

**Background:** Intranasal corticosteroids (INCS) are used for several conditions, including allergic rhinitis and sinusitis. Consequently, their safety profile needs continuous monitoring. This study aimed to analyse the adverse drug reactions (ADRs) of the INCS with market authorisation in Europe. **Methods:** A retrospective analysis of the ADR data from EudraVigilance in the period between the 1 January 2011 and 12 June 2023 was performed, with 2641 reports selected for analysis. ADRs were categorised by seriousness and evolution, with a focus on the most prevalent ADRs and their alignment with the Summary of Product Characteristics (SmPC). **Results:** The 18–64 age group, particularly females, was most affected. Mometasone was the most reported INCS, with predominantly non-serious ADRs, evolving towards resolution, which often were not listed in the SmPC. From the reported ADRs that were not described in the SmPC of the INCS studied, “Anosmia” and “Ageusia” were highlighted. Regarding the ADRs described in the SmPC, the most frequently reported were “Epistaxis” and “Headache”. The ADRs belonging to the Designated Medical Event list were also analysed, with “Angioedema” as the most reported ADR, which was mainly associated with budesonide. **Conclusions:** These findings underscore the importance of continuous INCS monitoring to mitigate ADRs and safeguard public health. Further research is warranted to explore potential novel signs for safety arising from previously unreported ADRs.

## 1. Introduction

Despite the fact that corticosteroids have a long history in medicine, significant advances in their clinical use and understanding of their action have been continuously observed over the years [1]. Various corticosteroids are produced naturally by the human body, and are essential hormones regulating countless physiological processes such as growth, metabolism, immunological response, inflammation, and response to stress [2,3]. Corticosteroids can be divided into two main groups: glucocorticoids and mineralocorticoids, which have different functions in the human body [4,5]. Of these, the most used are glucocorticoids, which are clinically applied in a wide range of situations including as anti-inflammatory and immunomodulatory agents [2,6]. Therefore, various formulations with glucocorticoids have been developed, accounting for the pathology of the disease being treated and the route of administration. In fact, glucocorticoids can be used orally, topically, inhaled, and injected [3]. Depending on the therapeutic purpose for glucocorticoids use, there are a variety of relevant aspects to be considered, including their adverse effects. Despite their high therapeutic potential, these drugs can also negatively affect various parts/functions of the body, namely, inducing *Diabetes mellitus* and/or depressive symptoms, particularly with prolonged use. Therefore, it is necessary to carefully consider the type of illness to be treated, the dosage, the most appropriate formulation as well as the risk/benefit ratio, while taking the potential adverse reactions into account. For example, some formulations can lead to more adverse effects than others, often depending on the achieved local/systemic levels of drugs. In this context, it is worth mentioning that oral and injectable glucocorticoid formulations, similar to the majority of other drugs, can induce more adverse drug reactions (ADRs) when compared to topical and inhaled formulations, which have a more localised effect [4,5]. The inhaled formulations with corticosteroids are frequently used for the control of respiratory pathologies such as asthma and chronic obstructive pulmonary disease (COPD) [4,5]. In turn, the intranasal formulations are often used for the relief of symptoms or the treatment of a wide range of nasal pathologies such as obstruction, sneezing, itching, rhinorrhoea, allergic rhinitis (AR), or sinusitis [7].

A relevant group of drugs administered by nasal topical application are the intranasal corticosteroids (INCS), which can be used as a primary therapy or as adjuvant drugs for many inflammatory conditions of the nasal cavity, including chronic sinusitis and AR [8,9]. Although INCS are considered relatively safe and efficient, there are variations in their potency, physicochemical properties (e.g., aqueous solubility), and pharmacokinetics, which can result in discrepancies in efficacy and safety [10]. Despite these differences, INCS are accepted as a front line for the treatment of pathologies that affect the nasal mucosa. In fact, these drugs can lead to the desired therapeutic effect with low doses because they are applied to the local area of action. The advantage of this type of topical action, in comparison with other oral and inhaling methods, is a lower level of systemic exposure and, consequently, less systemic adverse reactions. Therefore, the reported ADRs associated with INCS are mainly limited to the nasal mucosa, although systemic ADRs can also occur [7,9,10,11,12]. As local adverse reactions are light, transitional, and self-limited, it is normally not necessary to interrupt the course of therapeutics. Examples of more common and lighter local ADRs associated with INCS include epistaxis, irritation, dry nasal mucosa, and the feeling of burning and stinging [11]. More serious local ADRs are related to the atrophy or ulceration of the nasal mucosa and perforation of the nasal septum; however, the occurrence of this last ADR has been more associated with the administration technique [13]. Some examples of systemic ADRs that are associated with INCS include the suppression of the hypothalamus-pituitary-adrenal (HPA) axis, Cushing syndrome, hyperactivity, anxiety, depression, aggressiveness, osteoporosis, increase in intraocular pressure, and cataracts [7,9]. The INCS suppression effects on the HPA axis can, for example, influence child development. In fact, there are INCS drugs, such as beclomethasone, that significantly affect child development, slowing it down. However, there are others, namely mometasone, for which there is no evidence of this ADR, and this is why this drug is considered safer to administer to the paediatric population [9,10,12].

Pharmacovigilance, or drug safety monitoring, encompasses the science and activities related to the detection, assessment, understanding, and prevention of adverse effects and other drug-related problems. The European Union (EU) has established a rigorous system to evaluate the drug safety after market authorisation, which aims to take appropriate actions, when necessary, to protect public health. Therefore, this plays a central role in monitoring drug safety within the EU. To implement this, the European Medicines Agency developed EudraVigilance, a system designed for the reporting of suspected adverse effects. This system allows for the detection of signs regarding previously unknown adverse effects as well as new information related to known adverse effects [14,15].

The primary objective of this study was to collect data and comprehensively analyse the ADRs associated with the INCS authorised for the market in Europe. Specifically, this analysis focused on six commonly used INCS: beclomethasone, budesonide, flunisolide, fluticasone, mometasone, and triamcinolone. By examining real-world data reported to EudraVigilance, the centralised European database for monitoring suspected adverse reactions to drugs within the European Economic Area (EEA), this study aimed to provide valuable insights into the safety profile of these drugs in the general population. Through this analysis, we seek to contribute to the ongoing efforts to enhance patient safety by identifying and understanding potential adverse reactions associated with the use of INCS.

## 2. Materials and Methods

A retrospective comprehensive analysis of the ADRs associated with the suspected INCS drugs authorised for the market in Europe, used by the general population with no age restriction and reported to EudraVigilance between the 1 January 2011 and 12 June 2023, was performed. This information was collected from EudraVigilance’s database, accessible to the public on the web page https://www.adrreports.eu/pt (accessed on 13 June 2023) [14]. This website was set up in 2012 by the European Medicines Agency aiming to provide the public with access to reports of suspected ADRs. These reports are submitted electronically to the EudraVigilance database by national regulatory authorities and pharmaceutical companies that have marketing authorisation for the medicinal products, and are validated by pharmacovigilance specialists. These reports are used to assess the benefits and risks of medicinal products during their development and to monitor their safety once they have been authorised for use within the EEA. Their aim is to improve public health by supporting the safety monitoring of medicinal products and increasing transparency for stakeholders, including the public [14]. The INCS analysed were beclomethasone, budesonide, flunisolide, fluticasone, mometasone, and triamcinolone. The variables considered in the characterisation of the reports were as follows: the type of reporter (whether he/she is a health professional or not), the patient’s profile (age group and sex), the seriousness of each report and seriousness criterion, evolution of the reaction, and if the ADR was described or not in the Summary of Product Characteristic (SmPC) of the suspected INCS.

A deep analysis of each INCS as a suspect drug was individually performed using its International Nonproprietary Name (INN), and all of the available information about the reported cases was collected per year.

It is essential to highlight that each report corresponds to one patient, although each report can have more than one reported ADR. Firstly, 34,525 reports were collected. However, after filtering by the active substance, focusing only on reports where the corticosteroids were the only suspected drugs, and where the formulation was intranasal, 2641 reports were identified and analysed.

During the data analysis, several reports with incomplete data were identified—in these cases, the variable in question was described as “not specific” or “unknown”. In addition, some reports included more than one ADR, and for some ADRs, more than one seriousness criterion—in these cases, where the most serious criterion overlapped with the others, the less serious was disregarded. In the evolution analysis, there were also reports which had more than one ADR and, consequently, more than one evolution. In these cases, the most uncertain evolution was considered. The seriousness criterion was divided into: “Death”, “Life Risk”, “Hospitalisation”, “Incapacity”, “Congenital Anomaly”, or “Clinically Important Condition”, according to the European guidelines [15].

Characterization of the more prevalent ADRs by INCS, with terms belonging to the DME (Designated Medical Event) list, was also performed.

The statistical analysis of the data was carried out using Microsoft Office Excel 365. The data obtained were organised according to the variables studied and are presented in the appropriate tables and graphs.

## 3. Results

### 3.1. The Quantification of Notifications over the Years

By analysing the annual trend of reports between 2011 and 2023 (Figure 1), it is clear that a consistent increase of reports occurred until 2019, the year with the highest number (342). However, from this year onwards, the number of reports decreased. Until the 12 June 2023, the data obtained in 2023 are the lowest since the beginning of the study.

### 3.2. Population Characterisation Regarding the Age Group, Gender, and Type of Reporter

Concerning the characterisation of the population, the ages were divided into eight categories (0–1 months, 2 months–2 years, 3–11 years, 12–17 years, 18–64 years, 65–85 years, and not specified) and the sex was classified as male, female, and unknown.

The most affected age group was between 18–64 years with 1447 cases reported, followed by the non-specified group (487 cases), and then the 65–85 years age group with 470 cases (Figure 2).

The most affected sex was female with 57.48%, corresponding to 1518 cases reported from a total of 2641. Males were described in 1051 reports (39.79%), and in 72 reports (2.73%), the gender was not specified.

Regarding the reporter, in a total of 2641 cases reported to EudraVigilance, non-health professionals reported 1468, which corresponded to 55.59% of the total ADRs reported, whereas health professionals reported 1173 cases, corresponding to 44.41%.

### 3.3. Reports by Suspect Intranasal Corticosteroids

The most reported INCS were mometasone and fluticasone, with a total of 999 and 920 reports, respectively (Figure 3), followed by beclomethasone (298 reports) and budesonide (280 reports).

### 3.4. Adverse Reaction Characterisation

#### 3.4.1. The Distribution of Adverse Reactions by Seriousness and by Seriousness Criteria

In the total number of reports analysed, 1441 (54.56%) were considered “Not serious” and the remaining 1200 reports (45.44%) were “Serious”. Concerning the seriousness criteria, the criterion “Clinically Important” stood out from all the others, with 845 notifications, followed by “Hospitalisation” and “Incapacity” with 186 and 102 notifications, respectively, “Life Risk” with 37 reports, and “Death” with 19 reports. Finally, the “Congenital Anomaly” criterion appeared in two serious reports.

#### 3.4.2. Characterisation of the Clinical Status of the Patients Regarding the Evolution of Adverse Reactions

Regarding the clinical status of the patients, the majority recovered fully. In fact, the evolution classified as “cure” was the most reported (1116 reports, 42.26%), whilst “cure with sequelae” occurred in 35 patients (1.32%), and “non recovered” happened in 496 patients (18.78%). In 19 (0.72%) reports, the evolution was “death”, and in 975 (36.92%), the evolution was “unknown”.

#### 3.4.3. The Relationship between Seriousness and the Intranasal Corticosteroid, Age Group, and Gender

For the majority of the suspected INCS, the number of “Not Serious” reports was higher than “Serious” reports, with emphasis on mometasone. For beclomethasone and budesonide, the difference between the number of “Not Serious” vs. “Serious” reports was lower. Conversely, there were more serious reports for fluticasone and triamcinolone (Figure 4).

Concerning the relationship between the age group and seriousness, a total of 1200 cases were reported as “Serious” and 1441 as “Not Serious”. For example, in the “Not Specified” age group, 259 “Serious” cases were reported. On the other hand, for the 18–64 age group, in a total of 1447 reports, there was a higher number of “Not Serious” cases reported (858 cases) (Figure 5).

Regarding the relationship between seriousness and sex, the number of reports with the classification “Not Serious” was higher for both sexes, with a greater number for the female sex (Figure 6).

#### 3.4.4. The Distribution of Adverse Reactions Described in the Summary of Product Characteristics

The total number of ADRs reported was 6807. Of this total, 2696 (39.61%) were described in the Summary of Product Characteristics (SmPC), while the large majority of these ADRs (4111, 60.39%) were not described.

In Table 1, the six most reported ADRs of the INCS, as well as their frequency described in the SmPC, are presented. “Epistaxis” was the most reported ADR, with 319 reports, followed by “Headache” with 266.

Considering the ADRs not described in the INCS SmPC, the most reported was “Anosmia”, followed by “Ageusia”, with 107 and 62 reports respectively. The terms “Drug ineffective”, “Off label use”, and “Rhinorrhoea” were also significantly reported, with 62, 60, and 59 reports, respectively.

#### 3.4.5. Characterisation of the Most Prevalent Adverse Reactions with Terms Belonging to the Designated Medical Event List

The ADRs associated with these INCS and belonging to the DME list are presented in Figure 7. In the set of the six INCS analysed, a total of 12 terms were detected. Of these, “Angioedema” was the most prominent ADR with twenty-four reports, followed by “Blindness” and “Anaphylactoid reaction”, both with nine reports, and “Anaphylactic” reaction with eight reports.

In Table 2, ADRs belonging to the DME list that occurred for the different INCS and their corresponding frequencies are shown in a more detailed manner. For beclomethasone, “Anaphylactic Reaction” was the most reported DME term; for budesonide and fluticasone, the most reported term was “Angioedema”; the most reported term for mometasone was “Anaphylactoid Reaction”; and for triamcinolone, the term was “Blindness”. On the other hand, for flunisolide, there were no ADRs found on the DME list.

#### 3.4.6. The Distribution of Adverse Reactions with Terms Belonging to the Designated Medical Event List Regarding Their Descriptions in the Summary of Product Characteristics

The ADRs present in the DME list, and whether they are described or not for the respective INCS, are presented in Table 3. As previously referenced, for flunisolide, no ADRs from the DME list were reported, and therefore it was excluded from the table. Out of the 12 reported ADRs, only “Angioedema” and “Anaphylactic reaction” are specifically described in the SmPC.

## 4. Discussion

This study allowed for the characterisation of the reported ADRs associated with the INCS currently in use within Europe (beclomethasone, budesonide, flunisolide, fluticasone, mometasone, and triamcinolone). The data were obtained through EudraVigilance’s database and the selected ADRs were reported in the period between the 1 January 2011 and 12 June 2023 [14].

Until the year 2019, the ADR reporting tendency increased, with a slight fall in 2016. However, from 2019 to June 2023, the number of reports had been decreasing (Figure 1). One potential explanation for this decrease is the dynamic nature of EudraVigilance as a ‘live’ database. In fact, this database undergoes constant updates and data revisions to ensure the provision of the highest quality information, thereby aiming for the utmost accuracy, and this can reduce the number of ADRs [16,17]. Another explanation can be the COVID-19 pandemic situation and the subsequent vaccination campaigns, which led to intensive monitoring of the ADRs associated with vaccines administered against SARS-CoV-2, and consequently, the potential lower attention given to the ADRs from other drugs classes [18]. Additionally, the misinformation and ignorance of the population on the subject, the thinking that only serious ADRs should be reported, the lack of culture and awareness by the reporter, the work overload which health professionals are subject to, as well as the absence of reported ADR evolution feedback, are all factors that may contribute to subnotification [19]. It is also relevant to highlight that the data from the year of 2023 were only analysed until June, which meant it was difficult to predict the number of ADRs at the end of the year. Moreover, the fact that INCS are drugs which have been commercialised since relatively long ago can lead patients and health professionals to consider them safe, therefore reporting less ADRs. Therefore, ADRs tend to initially be reported more frequently for a recently introduced drug. However, over time, as the drug is considered safer, the number of reports tends to decrease. This fact could explain the increase in reports until 2019, followed by a decline thereafter.

Regarding the type of reporter, as previously mentioned, ADRs can be reported by health professionals and non-health professionals. Health professionals constitute pharmacists, doctors, nurses, and other professionals working in this area, whereas non-health professionals constitute patients, families, and caregivers [15]. However, the data obtained did not specify the type of health professional. Despite this, in the analysis performed, it was verified that non-health professionals reported a higher number of ADRs. This provides evidence that the measure implemented for the general population to participate in the reporting of ADRs was positively accepted, leading to easier access to drug information. In this sense, the collaborative efforts of non-health professionals substantially contributed to the ADR notifications associated with INCS utilisation, thereby facilitating improvements in these medications’ safety profiles [20,21]. The higher number of reports from the non-health professional group is in opposition to what has been the usual tendency. Throughout the years, health professionals had the highest reporting rate, with the highest reporters being doctors, followed by pharmacists [22]. It is also important to mention that, in general, there are differences in reported information between ADR reports between patient and healthcare professionals. Patient reports tended to focus more on patient-related information and the impact of the reported ADRs, whereas healthcare professional reports provided more clinically relevant details [23].

Regarding the age group, the most reported group was between 18 and 64 years old (Figure 2), which constituted the predominant age group in Europe. Consequently, this was the group with more ADRs reported in different studies [24]. The most reported sex was female, which also matched reports described in the literature. According to some studies, women are more interested in and seek to find out more information about their own health than men. In addition, females generally use more medication [25]. Another hypothesis to explain this is the physiological difference between both sexes, for example, in hormone levels and metabolism. For instance, women have a lower glomerular and hepatic filtration rate than men, lower lean mass, but more adipose tissue. This set of factors will affect, namely, the activities of the complex P450 enzymes involved in the metabolism involved in the majority of pharmaceuticals. Thus, these and other physiological factors may explain why women have greater susceptibility to adverse reactions [25,26].

Considering all of the INCS studied, mometasone and fluticasone were the most reported as suspected drugs (Figure 3), perhaps because they can be used in children and adolescents. In this context, it is important to refer to evidence that corticosteroids have an adverse effect of causing growth suppression in children and adolescents. However, this situation was not evident for mometasone and fluticasone, as these drugs are considered safer, and thus are approved for use by this population. Despite this, it has been described that some doctors are reluctant to prescribe these drugs to these age groups [8,10,12].

Most of the reports were “Not Serious” (54.56%), contrary to several studies that have demonstrated that serious reactions are more often reported. In fact, reporters generally consider that only safe drugs are commercialised [27,28]. However, other studies have demonstrated that “Not Serious” reactions have been gaining relevance and are being increasingly taken into consideration to improve the drugs’ security profile [29]. The most reported INCS in these cases was mometasone (Figure 4). Contrarily, more serious reports were present for fluticasone despite various studies providing evidence that the benefit overcame the risks in paediatric ages [10,12,30]. Considering the seriousness, the criterion “Clinically Important” was notoriously dominant over all of the others, which corroborated another study reporting the same incidence of this seriousness criterion [15,31]. In the present study, most patients recovered fully from their ADRs, which is expected, taking into consideration the higher number of “Not Serious” reports and the fact that the reactions associated with these drugs are considered in a general light and are self-limited [9].

An analysis of the reported ADRs described in the SmPC of the INCS was also performed. Interestingly, the majority (60.39%) of ADRs were not described in the SmPC. Of these, “Anosmia”, “Ageusia”, “Inefficient Medication”, “Off label use”, and “Rhinorrhoea” were highlighted. Considering the ADRs described in the SmPC (Table 1), the most reported were “Epistaxis”, “Headaches”, “Dyspnoea”, “Visual disturbances”, “Dizziness”, and “Blurred vision”. These results are similar to those obtained by Rollema et al. in their analysis of ADR reports associated with INCS, reported in the Dutch database of spontaneous reports from the Netherlands Pharmacovigilance Centre Lareb [29]. The reported ADRs described in the SmPC are also referred in various articles, even those that occur less frequently, such as “Suppression of the hypothalamic-pituitary-adrenal axis” and “Increase in the intraocular pressure” [8,32]. Although not occurring at a higher frequency when compared with the aforementioned reactions, “Nasal septum perforation” is an ADR described in the SmPC and also is a common ADR [32,33]. However, this reaction has been related to the possible incorrect use of the INCS device. Thus, this can be avoided if the health professional that accompanies the patient, namely the pharmacist, who at the time of dispensing the medication, instructs the patient in the correct use of the device [8,13]. Regarding the non-described ADRs, it is also important to highlight “Drug ineffective”. One possible hypothesis for the frequency of this reaction is the time required to observe positive effects by the INCS, because they only reach the maximum clinical efficiency after some days or weeks [7,13]. Considering this, the reports may have occurred before the expected effective time. It is of extreme importance in these situations that the health professionals intervene and explain that effects may not be immediate [34].

Regarding the terms belonging to the DME list, 64 ADRs, corresponding to 12 terms (Figure 7) belonging to the DME list, were analysed. The term which stood out most was “Angioedema”. This ADR was described in four of the six INCS analysed, and budesonide was the corticosteroid of this group with the highest incidence of this reaction. It is important to mention that this reaction appears in the SmPC of the INCS, including for budesonide [32,33]. The fact that not having the description of various DME terms highlights that new safety signs may be present, which deserves further investigation. However, due to the limitation of available information in the reports, it was not possible to produce a more detailed analysis of each report. In fact, there are missing clinical data that could allow for the exclusion of other comorbidities or other causes for the DME ADRs not described.

Despite the limitations of the obtained results, they may provide a real-world assessment of the safety profile of these drugs. One of the most important limitations of this study was the low rate of the already-known associated reporting. In fact, various studies address this difficulty, and some even suggest measures that can be included to increase this rate [35]. In addition, there were also reports with incomplete or confusing information regarding the ADRs, which made it difficult to produce adequate analysis, mainly due to the publicly available information being generally minor. Moreover, the study compared reports of multiple intranasal corticosteroids, with varying quotas across different countries within the EEA, and as such, the results may be affected by this situation. Another limitation was the limited temporal coverage. This study analysed data up to June 2023, which limits the ability to assess more recent trends and changes in the safety profile of intranasal corticosteroids after this date.

## 5. Conclusions

This study reflects the importance of pharmacovigilance and its contribution to the safety profile of medicines. Concerning the characterisation of INCS ADRs, females were more affected, as well as the 18 to 64 age group. Additionally, the drugs mometasone and fluticasone stood out in our analysis, with mometasone being slightly more reported as a suspected drug. However, the majority of ADRs were considered “Not Serious” and the patients recovered completely. Despite this, some ADRs, including “Anosmia”, “Ageusia”, and “Inefficient Medication”, were not described in the SmPC and thus deserve further detailed analysis. In view of these results, it was concluded that ADRs should continue to be monitored to protect public health, and that new studies should be carried out to confirm whether undescribed ADRs could be new safety signals. Future studies should focus on more detailed regional and demographic analyses to identify specific safety concerns among different populations and locations. This could help to better understand how different factors, such as age, sex, and health conditions affect the occurrence of ADRs.

## Figures and Tables

**Figure 1 healthcare-12-01923-f001:**
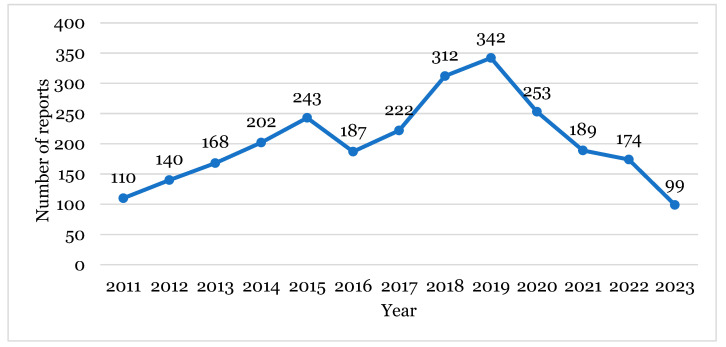
The number of reports per year, between 2011 and 2023.

**Figure 2 healthcare-12-01923-f002:**
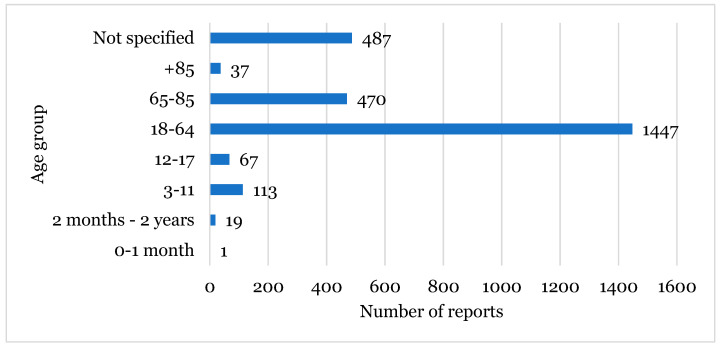
Population characterisation regarding the age group.

**Figure 3 healthcare-12-01923-f003:**
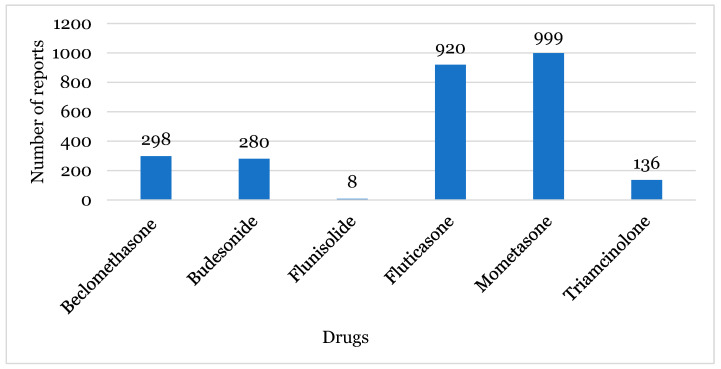
The number of reports by intranasal corticosteroids.

**Figure 4 healthcare-12-01923-f004:**
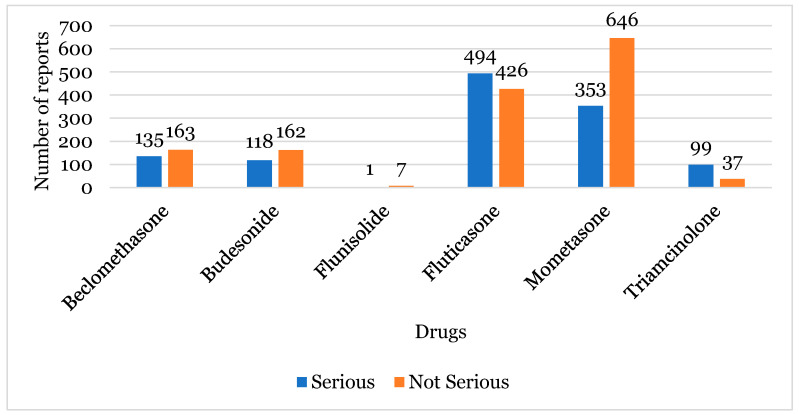
The relationship between the seriousness and the intranasal corticosteroid suspect.

**Figure 5 healthcare-12-01923-f005:**
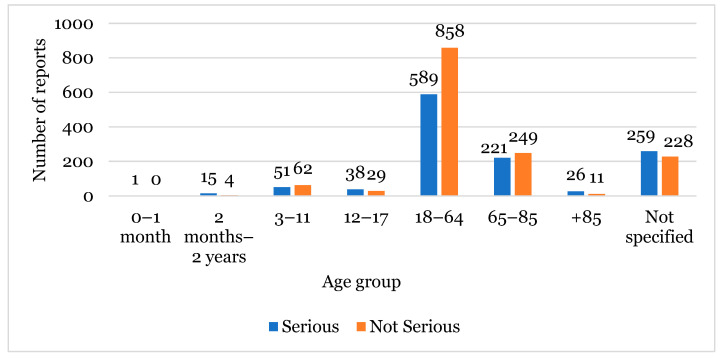
The relationship between seriousness and age group associated with adverse reactions.

**Figure 6 healthcare-12-01923-f006:**
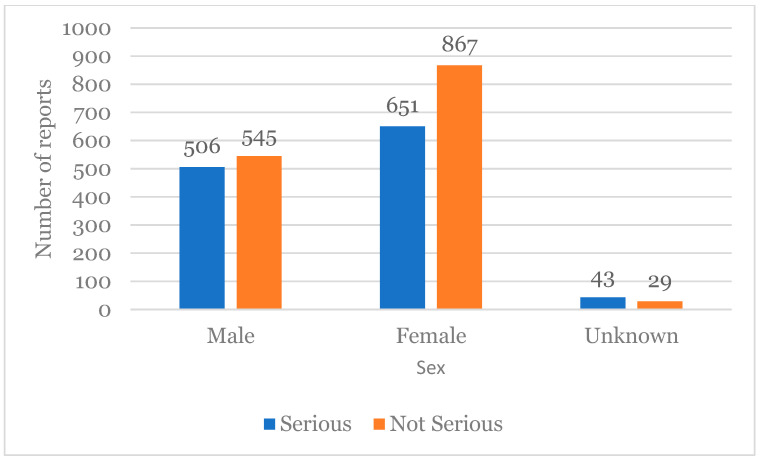
The relationship between seriousness and sex associated with adverse reactions.

**Figure 7 healthcare-12-01923-f007:**
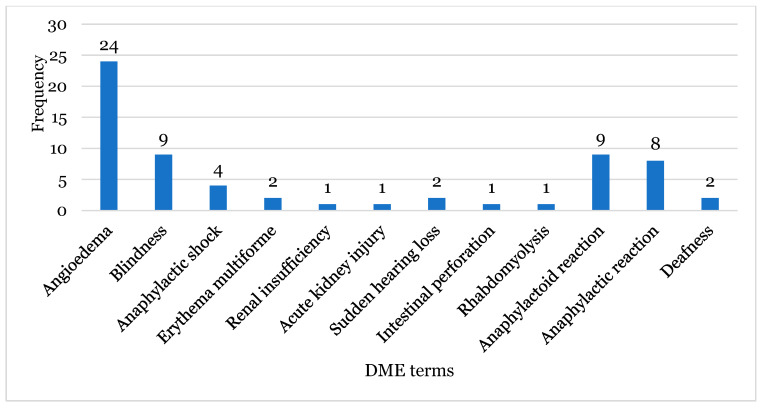
The frequency of the adverse drug reactions belonging to the Designated Medical Event (DME) list.

**Table 1 healthcare-12-01923-t001:** The number of reports for the 6 most reported adverse reactions, and their frequencies as described in the Summary of Product Characteristics (SmPC).

ADR ^1^	Number of Notifications	Frequency Described in the SmPC ^2^
Epistaxis	319	Very frequent
Headache	266	Frequent
Dyspnoea	143	Unknown
Visual disturbances	110	Unknown
Dizziness	103	Unknown
Blurred vision	103	Unknown

^1^ ADR: Adverse Drug Reaction; ^2^ SmPC: Summary of Product Characteristic.

**Table 2 healthcare-12-01923-t002:** The relationship between the terms of the Designated Medical Event list and the frequency of adverse drug reactions of each intranasal corticosteroid.

DME ^1^ Terms	Frequency of ADR ^2^ Incidence					
	Beclomethasone	Budesonide	Flunisolide	Fluticasone	Mometasone	Triamcinolone
Angioedema	1	12	0	7	4	0
Blindness	0	2	0	3	2	2
Anaphylactic shock	1	0	0	1	2	0
Erythema multiforme	0	0	0	2	0	0
Renal insufficiency	0	1	0	0	0	0
Acute kidney injury	1	0	0	0	0	0
Sudden hearing loss	0	0	0	0	2	0
Intestinal perforation	0	0	0	0	0	1
Rhabdomyolysis	0	0	0	1	0	0
Anaphylactoid reaction	0	1	0	0	8	0
Anaphylactic reaction	3	0	0	5	0	0
Deafness	1	0	0	1	0	0
Total	7	16	0	20	18	3

^1^ DME: Designated Medical Event; ^2^ ADR: Adverse Drug Reaction.

**Table 3 healthcare-12-01923-t003:** The distribution of adverse reactions found in the terms of the Designated Medical Event list and its description status in the Summary of Product Characteristics.

DME ^1^ List Terms	INCS ^2^	Description in the SmPC ^3^
Angioedema	Beclomethasone;	Described
	Budesonide;	
	Fluticasone;	
	Mometasone	
Blindness	Budesonide;	Not Described
	Fluticasone;	
	Mometasone;	
	Triamcinolone	
Anaphylactic shock	Beclomethasone;	Not Described
	Fluticasone;	
	Mometasone	
Erythema multiforme	Fluticasone	Not Described
Renal insufficiency	Budesonide	Not Described
Acute kidney injury	Beclomethasone	Not Described
Sudden hearing loss	Mometasone	Not Described
Intestinal perforation	Triamcinolone	Not Described
Rhabdomyolysis	Fluticasone	Not Described
Anaphylactoid reaction	Budesonide;	Not Described
	Mometasone	
Anaphylactic reaction	Beclomethasone;	Described
	Fluticasone	
Deafness	Beclomethasone;	Not Described
	Fluticasone	

^1^ DME: Designated Medical Event; ^2^ INCS: Intranasal Corticosteroid; ^3^ SmPC: Summary of Product Characteristics.

## Data Availability

All data that the authors extracted from the European Medicines Agency (EMA) pharmacovigilance database, called EudraVigilance, are publicly available. Pharmacovigilance data from EudraVigilance are publicly available at www.adrreports.eu (accessed on 14 June 2023).

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
