# Peer review of "Risks Associated with the Use of Intranasal Corticosteroids: An Analysis of Adverse Reactions Reported to EudraVigilance"

_healthcare, 2024, doi:10.3390/healthcare12191923_

Round 1

Reviewer 1 Report

Comments and Suggestions for Authors

Major comments:

This paper focuses on intranasal corticosteroids, which, while commonly used in daily clinical practice, are often prescribed without much caution due to the non-severe diseases they treat. The study presents real-world data on the side effects of these medications, serving as a valuable reminder to prescribing physicians to exercise caution. It also raises awareness among patients about the risks of casual use, making this paper clinically significant.

However, there are some concerns and questions:

・Given that many readers may be outside of Europe, it would be beneficial to include an explanation of the EudraVigilance registration process in the introduction or methods section.

・Has the difference in the characteristics of adverse events reported by health professionals versus non-health professionals been considered?

・The comparison of multiple intranasal corticosteroids may not be fair without understanding the differences in market share in Europe. Additionally, if there are newly approved medications, this could also impact the results. It would be advisable to address these points.

・Why has the number of reported cases increased until 2019?

Minor comment:

・In Figure 7, there is a part where the name of the adverse event on the horizontal axis is cut off.

Author Response

This paper focuses on intranasal corticosteroids, which, while commonly used in daily clinical practice, are often prescribed without much caution due to the non-severe diseases they treat. The study presents real-world data on the side effects of these medications, serving as a valuable reminder to prescribing physicians to exercise caution. It also raises awareness among patients about the risks of casual use, making this paper clinically significant.

Author´s response: Thank you very much for your comments and interest in our work. The manuscript was now revised according to the comments received, which allowed the quality of this article to improve. Please see the revised manuscript version.

However, there are some concerns and questions:

・Given that many readers may be outside of Europe, it would be beneficial to include an explanation of the EudraVigilance registration process in the introduction or methods section.

Author´s response: Thank you very much for this observation. We recognize that some readers may be unfamiliar with EudraVigilance and therefore we provided a brief explanation in the Introduction and in the Materials and Methods sections. Please see the revised manuscript version (page 2, lines 89-95; page 3, lines 111-120)

・Has the difference in the characteristics of adverse events reported by health professionals versus non-health professionals been considered?

Author´s response: Thank you for your question. In fact, some studies evaluated the differences in the information reported between patient and healthcare professional. Patient reports tend to focus more on patient-related information and the impact of the reported ADRs, whereas healthcare professional reports provide more clinically relevant details. We added this information in the discussion section and added a new reference. Please see the revised manuscript version (page 11, lines 330-334)

Rolfes L, van Hunsel F, Wilkes S, van Grootheest K, van Puijenbroek E. Adverse drug reaction reports of patients and healthcare professionals-differences in reported information. Pharmacoepidemiol Drug Saf. 2015 Feb;24(2):152-8. doi: 10.1002/pds.3687. Epub 2014 Jul 31. PMID: 25079444.

・The comparison of multiple intranasal corticosteroids may not be fair without understanding the differences in market share in Europe. Additionally, if there are newly approved medications, this could also impact the results. It would be advisable to address these points.

Author´s response: Thank you very much for this advice. We agree with you that reports of multiple intranasal corticosteroids, with varying quotas across different countries within the EEA, could affect the results and we added this note at the discussion section. Please see the revised manuscript version (page 12-13, lines414-416)

・Why has the number of reported cases increased until 2019?

Author´s response: Thank you very much for your comment. When a new drug is launched on the market, healthcare professionals usually show increased vigilance in reporting ADRs. This increased vigilance is probably due to the limited clinical experience and uncertainty about the safety profile of the drug in the general population. The longer the drug is on the market and the more it is used, the more confidence healthcare providers develop in the safety of the drug. This change in perception may lead to a decrease in adverse event reporting over time as the drug is seen as more established and less likely to cause unexpected side effects. This phenomenon may explain the initial increase in ADR reporting, followed by a gradual decline in reporting in subsequent years. We have addressed this issue in the discussion section. Please see the revised manuscript version (page 10-11 , lines311-314)

Minor comment:

・In Figure 7, there is a part where the name of the adverse event on the horizontal axis is cut off.

Author´s response: Thank you very much for your comments. We changed the figure 7. Please see the revised manuscript version

Reviewer 2 Report

Comments and Suggestions for Authors

The authors conducted a very interesting retrospective study by evaluating the adverse drug reactions reported with intranasal corticosteroids from year 2011 till 2023.i have a few comments needs clarification.

1.Please add a paragraph on pharmacoviglince in the introduction of the manuscript you missed.

2.In material and method section you didn't specifically stated the methods of collection of the data.

3.Was the data attuntentic if yes who reported these adverse drug reactions.

4.please add the limitations and the future perspectives of you study at the end

5.check for minor spelling mistakes

Comments on the Quality of English Language

Recheck the manuscript for minor typographical mistakes 

Author Response

The authors conducted a very interesting retrospective study by evaluating the adverse drug reactions reported with intranasal corticosteroids from year 2011 till 2023.i have a few comments needs clarification.

Author´s response: Thank you very much for your comments and interest in our work. The manuscript was now revised according to the comments received, which allowed the quality of this article to improve. Please see the revised manuscript version.

1.Please add a paragraph on pharmacoviglince in the introduction of the manuscript you missed.

Author´s response: Thank you very much for your remarks. We agree with you that a brief reference to the topic of pharmacovigilance could enhance the introduction, and we have added this information in the Introduction section as per your suggestion. Please see the revised manuscript version (page 2, lines 87-95)

2.In material and method section you didn't specifically stated the methods of collection of the data.

Author´s response: Thank you for your comment. As mentioned in the article, the data was extracted from the website https://www.adrreports.eu/pt. This website was set up in 2012 by the European Medicines Agency with the aim of providing the public with access to reports of suspected ADRs. These reports are submitted electronically to the EudraVigilance data-base by national regulatory authorities and pharmaceutical companies that have marketing authorization for the medicinal products. The reports are used to assess the benefits and risks of medicinal products during their development and to monitor their safety once they have been authorised in the EEA. The aim is to improve public health by supporting the safety monitoring of medicinal products and increasing transparency for stakeholders, including the public. We added this information in the methods section for a better understanding. Please see the revised manuscript version (page 3, lines 111-120)

3.Was the data attuntentic if yes who reported these adverse drug reactions.

Author´s response: Thank you very much for your questions. Regarding this matter, we would like to clarify that all information related to suspected adverse effects available on the website https://www.adrreports.eu/pt comes from EudraVigilance, a database designed for collecting reports of suspected adverse effects. This database can be used to assess the benefits and risks of medicines during their development and to monitor their safety following market authorization within the EEA. The data included in EudraVigilance is electronically submitted by national regulatory authorities and pharmaceutical companies holding marketing authorizations for the medicines, and is duly validated by pharmacovigilance specialists. We provided a brief explanation in the Materials and Methods section. Please see the revised manuscript version (page 3 , lines 111-120)

4.please add the limitations and the future perspectives of you study at the end

Author´s response: Thank you very much for your suggestion. In addition to the limitations already described in the discussion section, we have added the following limitations:  “Moreover, the study compared reports of multiple intranasal corticosteroids, with varying quotas across different countries within the EEA, and as such, the results may be affected by this situation. Another limitation was the limited temporal coverage. In fact, the study analysed data up to June 2023, which limits the ability to assess more recent trends and changes in the safety profile of intranasal corticosteroids after this date.”.

As for future perspectives, we added in Conclusions that, in addition to the continuous monitoring of adverse effects of intranasal corticosteroids, future studies should focus on more detailed regional and demographic analyses to identify specific safety patterns among different populations and locations. This could help to better understand how different factors, such as age, sex and health conditions, affect the occurrence of adverse events.

Please see the revised manuscript version (page 13 , lines 410-415; lines 426-429)

5.check for minor spelling mistakes

Author´s response: Thank you very much for your advice. Following your suggestion, the document has been reviewed again, please see the revised manuscript version

Reviewer 3 Report

Comments and Suggestions for Authors

Dear Author(s)

The submitted manuscript is a good collection of data. However, it will be more accurate if the dose and duration of treatment are also included in the study. It's unclear after how many days of treatment the reported serious and non-serious adverse effects were observed in the manuscript (in the abstract, keywords, discussion and section).

Author Response

Dear Author(s)

The submitted manuscript is a good collection of data. However, it will be more accurate if the dose and duration of treatment are also included in the study. It's unclear after how many days of treatment the reported serious and non-serious adverse effects were observed in the manuscript (in the abstract, keywords, discussion and section).

Author´s response: Thank you very much for your comments and interest in our work. We agree with you that incorporating dose and duration of treatment could improve both the results and their interpretation. However, as mentioned in the discussion, due to the limitations of the available information in the reports, a more detailed analysis of each report was not feasible. In particular, there is a lack of data on dose and treatment duration, as well as clinical information that could help exclude other comorbidities or causes of adverse drug reactions (ADRs) not described. The information available on the website https://www.adrreports.eu/pt complies with the criteria outlined in the EudraVigilance Access Policy. The online notifications include aggregated information related to suspected adverse effects, using data elements from reports submitted to EudraVigilance. The lists provide an overview of individual cases in a tabular format, while the safety notification forms for individual cases present specific case details; however, we did not have access to these forms. Despite the limitations of the obtained results, they may still offer a valuable real-world assessment of the safety profile of these drugs

Round 2

Reviewer 1 Report

Comments and Suggestions for Authors

I believe this manuscript has been well revised based on the points I raised. I have no further comments.